# Inhaled Antibiotics for Mycobacterial Lung Disease

**DOI:** 10.3390/pharmaceutics11070352

**Published:** 2019-07-19

**Authors:** Brandon Banaschewski, Thomas Hofmann

**Affiliations:** Department of Research and Development, Qrumpharma Inc., Doylestown, PA 18901, USA

**Keywords:** inhalation, antimicrobials, tuberculosis, nontuberculous mycobacteria

## Abstract

Mycobacterial lung diseases are an increasing global health concern. Tuberculosis and nontuberculous mycobacteria differ in disease severity, epidemiology, and treatment strategies, but there are also a number of similarities. Pathophysiology and disease progression appear to be relatively similar between these two clinical diagnoses, and as a result these difficult to treat pulmonary infections often require similarly extensive treatment durations of multiple systemic drugs. In an effort to improve treatment outcomes for all mycobacterial lung diseases, a significant body of research has investigated the use of inhaled antibiotics. This review discusses previous research into inhaled development programs, as well as ongoing research of inhaled therapies for both nontuberculous mycobacterial lung disease, and tuberculosis. Due to the similarities between the causative agents, this review will also discuss the potential cross-fertilization of development programs between these similar-yet-different diseases. Finally, we will discuss some of the perceived difficulties in developing a clinically utilized inhaled antibiotic for mycobacterial diseases, and potential arguments in favor of the approach.

## 1. Introduction

*Mycobacterium* is a genus of Actinobacteria that includes roughly 190 bacterial species which are characterized by a waxy, mycolic acid rich cell wall that imparts resistance to osmotic pressure, environmental factors, and antibiotics [1]. Pathogenic mycobacteria are known for their ability to form biofilms, as well as invasion of and survival within host cells [2]. As such, clinical *Mycobacterium* infections are difficult to treat and require intensive multidrug treatment strategies lasting months or years [3,4,5].

In terms of mycobacterial lung infections, there are two main etiological agents, *Mycobacterium tuberculosis* and nontuberculous mycobacteria (NTM). *Mycobacterium tuberculosis*, the causative agent of tuberculosis (TB), remains one of the most common causes of infection and mortality worldwide. Global estimates suggest that in 2017, there were 10 million infections and 1.3 million TB-related deaths [5]. Comparatively, NTM lung disease is less well understood, although it is slowly being recognized as a major global health concern due to its steadily increasing prevalence worldwide [6,7]. Unlike *M. tuberculosis* complex, NTM are opportunistic pathogens that often infect patients with underlying conditions, such as bronchiectasis, chronic obstructive pulmonary disease, and cystic fibrosis, although disease can sometimes occur idiopathically [8]. While the incidence of TB has begun to slowly decline (2.3% reduction from 2016), the global prevalence of NTM pulmonary disease has rapidly increased [5,9,10,11,12,13,14]. Even more concerning is the growing realization that the prevalence of NTM worldwide may even be larger than estimated due to misdiagnosis [7,15,16].

Mycobacterial lung infections are generally caused by inhalation of aerosols. The source of these aerosols may be environmental, as is frequently the case for NTM, or from other infected individuals, as is noted for TB. Once in the lungs, the pathophysiology of these infections is believed to be relatively similar, although clinical severity appears to vary [8,17,18,19]. In both cases, the invading pathogens are quickly recognized and phagocytosed by alveolar macrophages, where the mycobacteria survive and proliferate intracellularly. In response, the body recruits a number of circulating monocytes, neutrophils, T-cells, and dendritic cells to form a granuloma, one of the hallmarks of mycobacterial lung infections [20,21]. This strategy often permits the survival of pathogens within the quarantined area leading to tissue cavitation, dissemination, and respiratory function decline [22]. Therefore, any mycobacterial therapy needs to be able to penetrate this inflammatory milieu to effectively target the invading pathogens.

Despite these pathophysiological similarities, clinical disease presentation differs between TB and NTM disease. Active TB is often much more severe, with the formation of numerous lung cavities and high mortality rates [18]. On the other hand, NTM disease is variable, and clinical presentation varies between milder bronchiectatic phenotypes, to tuberculosis-like fibrocavitary disease [3,23].

### 1.1. Current Treatment Strategies and Limitations

Treatments for both TB and NTM diseases are often long and rigorous, mainly due to the arduous nature of mycobacteria and the development of granulomatous structures. Current guidance-based therapies (GBT) for TB were developed in the 1960–1970s, and involve administration of isoniazid, rifampicin, ethambutol, and pyrazinamide for 6–30 months, with success rates typically ranging between 76% and 94% for drug susceptible infections [3,5]. However, major concerns in the treatment of TB are the development of multidrug resistant (MDR) (which is defined as resistance to isoniazid and rifampicin), and/or extensively drug resistant (XDR) strains (defined as bacterial resistance to isoniazid, rifampicin, a fluoroquinolone, and at least one second-line injectable drug, i.e., capreomycin, kanamycin, and/or amikacin) [24]. These drug-resistant infections were responsible for 600,000 new infections in 2017, and resulted in 240,000 deaths worldwide [5]. Treatment success in these drug-resistant infections is significantly lower, with a success rate of 54% in MDR-TB patients, and a mere 30% for patients with XDR-TB [5]. For these drug resistant infections, treatment regimens become varied, and treatment durations can last as long as 24 months. These regimens are toxic, poorly tolerated, and often based on empirical data alone [25].

Unlike TB, treatment of NTM infections is severely understudied, and prospective clinical trials to identify ideal antibiotic regimens have been lacking. As such, current therapeutic guidelines draw heavily on regimens for TB [3]. There are general therapeutic principles that are uniformly followed across all pathogenic NTM species, such as the use of three to four antibiotics, and that therapy should continue for at least 12 months following sputum conversion [3]. After these basic principles, treatment regimens vary depending on species, clinical phenotype, and drug susceptibility profiles [26,27,28], leading to favorable treatment outcomes ranging between 8–88% [29,30,31,32,33]. The variability in successful treatments are often predicated upon the causative agents (i.e., *M. avium* versus *M. abscessus*), bacterial subspecies (i.e., *M. abscessus* subsp. *massiliense* versus *M. abscessus* subsp. *abscessus*), disease phenotype (i.e., nodular bronchiectatic versus fibrocavitary), and treatment regimens employed (i.e., macrolide-containing regimens versus those without macrolide use). Patient survival following NTM infection is similarly variable: median survival of patients with *M. avium* complex infection was 13.0 years, while median survival time following infection with other NTM species was only 4.6 years [34].

Treatment of NTM-PD often leads to a significant economic burden for patients due to a combination of long drug therapy regimens, frequent medical examinations, and the risk of hospitalization. In the United States, therapy has been estimated to cost $19,876–$37,579 per patient, and cost an estimated $815 million in 2010 alone [11,35,36].

Even though treatment success rates of drug-susceptible mycobacterial infections are high, discontinuation rates are common, ranging from 7–53.6% in patients with drug susceptible TB [37], and 9–39% in NTM disease patients [38,39,40,41]. The major causes for patient discontinuation are long treatment durations, lack of observed improvement, and the severe side effects associated with oral and parenteral dosing [23,38,42,43,44]. Common adverse events include gastrointestinal intolerance, neuropathy, myelotoxicity, and skin rashes, but there are more severe events, such as renal failure, ototoxicity, and hepatotoxicity [18,25]. Any premature cessation of treatment dramatically increases the risk of developing MDR infections, and supportive measures are often required to increase adherence, although the success of these measures is controversial [45].

Because of the long, difficult-to-treat nature of mycobacterial lung infections, aerosol drug delivery would be an ideal strategy to specifically target the infection site. Inhaled drug delivery has repeatedly been shown to increase drug concentrations within the lungs, while minimizing extrapulmonary exposure. This targeted delivery has been extensively reviewed, and studies consistently demonstrate better drug tolerance compared to systemic administrations, and improved clinical outcomes over the course of chronic therapy [46].

### 1.2. The Potential for Inhaled Therapies

In an attempt to develop new, more effective therapies for both drug-susceptible and drug-resistant mycobacterial infections, several groups have turned their attention towards the development of inhaled antibiotics. The successful development of pulmonary therapies for other indications, such as asthma, chronic obstructive pulmonary disease, and other chronic pulmonary infections (particularly *Pseudomonas aeruginosa* in cystic fibrosis patients [47,48,49]), has demonstrated the validity of this approach. The administration of inhaled therapeutics has been repeatedly shown to increase therapeutic drug concentrations within the lungs, and reduce the incidence of off-target side effects by limiting overall systemic exposure [46].

Additionally, the ability of mycobacteria to reside and proliferate within alveolar macrophages renders drug uptake by macrophages a crucial step for effective antimicrobial activity. Numerous studies have demonstrated that inhaled particles or vesicles are readily engulfed by these alveolar macrophages, and that inhaled drug delivery may enhance anti-mycobacterial activity [50]. This mechanism of uptake may help treatment outcomes by increasing therapeutic concentrations within infected cells and granulomas, thereby improving drug efficacy.

In this review, we will discuss past studies investigating aerosol treatment strategies, presently available inhaled therapeutics for the treatment of both NTM and TB, and novel concepts that are currently in development to become first-line therapies for the treatment of mycobacterial lung disease.

## 2. Drug Delivery Strategies for Inhaled Antibiotics

Delivery of antibiotics to the lungs requires use of an aerosol drug delivery device to disperse the drugs into particles suitable for inhalation. Antibiotic therapy for the treatment of pulmonary infections requires delivering relatively large doses of medication to the lungs. As an example, TOBI^®^ (tobramycin inhalation solution) requires a 300 mg dose of for management of *Pseudomonas aeruginosa* infections, whereas administration of Ventolin^®^, albuterol sulfate inhalation powder for the treatment of bronchospasm, requires a dose of only 108 µg. The antibiotic dose is more than 1000 times larger than the bronchodilator medication.

Traditionally, the large doses required for antimicrobial therapy have been delivered to the lungs via nebulizer device. Here, the drug is formulated in a solution or suspension, and is placed in the nebulizer immediately before use. Patients place the mouthpiece of the nebulizer in their mouth and inhale the drug over a 5 to 20 min period [51]. The advantages of nebulizer treatment include easier formulation and manufacturing of the drug product, and nebulizers can be used by any patient, regardless of their pulmonary function [51]. The disadvantages include the up-front cost of the nebulizer device, requirement of a power supply, comparatively long treatment durations, and the bulky drug product that can be more expensive to transport than other inhalation products.

Recently, dry powder technology and powder formulation technology have developed that enable delivery of large doses suitable for antibiotic therapy. One example is the TOBI^®^ Podhaler, tobramycin inhalation powder, which can deliver 112 mg of drug and was approved for use in the US in 2013. TOBI^®^ Podhaler combines a capsule based inhaler with a PulmoSphere™ based formulation and delivers the required dose over four capsules [52]. Further, disposable devices have been developed that can deliver doses >50 mg in a single breath [53]. The advantages of a dry powder inhaler-based therapy are more convenient packaging, a simpler device, and shorter treatment durations. However, they have limited patient populations, as patients require adequate pulmonary function to inhale the antibiotic effectively. Furthermore, while a dry powder inhaler may appear simple, development is often more complex and riskier than nebulizer formulations [54,55]. As a result, in high dose applications, dry powder inhaler-based products are often more expensive than their nebulizer-based equivalents [56].

## 3. Past and Present Inhaled Therapies for Mycobacterial Lung Disease

### 3.1. Tuberculosis

The initial rationale for developing an inhaled therapy for the treatment of TB was the concern over antibiotic resistance (specifically streptomycin resistance) [50,57]. One of the first aerosol programs investigated the use of aerosol streptomycin for the treatment of TB in 12 children in order to increase the drug site concentration within the lungs and overcome any potential streptomycin resistance [58]. None of the patients showed any symptoms of toxicity following inhaled treatment of 2 g streptomycin daily, and nine of the children demonstrated “healing”, which was described as either a reduction in the size or elimination of granulomatous structures from the lungs. This was the first report to describe aerosol delivery for the treatment of TB lesions, and the positive results presented a promising avenue of investigation. However, over the next decade, development of isoniazid, pyrazinamide, ethambutol, and rifampicin shifted the focus away from inhaled drugs, as these drugs became the core antibiotics in TB treatment regimens.

Almost 50 years later, the emergence of MDR- and XDR-TB has again precipitated the need for novel therapeutics, and has renewed interest in inhaled therapeutics. With the clinical use of parenteral antibiotics for MDR- and XDR-TB, there is an added incentive to develop inhaled delivery over these injectable therapies, as patients frequently report pain at the injection sites [59] and the regimen is associated with high rates of discontinuation and non-adherence [60].

One of the major antimicrobial classes that have been investigated for MDR- and XDR-TB are aminoglycosides, which are conventionally used as second-line injectable drugs. In a small pilot study by Turner et al., five patients received inhaled kanamycin in addition to GBT for the treatment of MDR-TB. The drug was well tolerated and all patients achieved sputum conversion in less than 60 days [61]. Following these promising results, Sacks et al. investigated the addition of either inhaled gentamicin or kanamycin to GBT in refractory TB patients, and found that 68% of patients achieved sputum conversion, with a mean time-to-conversion of 33 days [62]. Since then, preclinical studies have investigated liposomal formulations of aminoglycosides, such as kanamycin [63], or fluoroquinolones, such as ofloxacin [64] and levofloxacin [65]; all which have demonstrated efficacy either in vitro or in vivo.

Further, multiple groups have investigated inhalable fixed dose combinations of first-line drugs [66]. The most commonly evaluated dry powder combination has been a rifamycin with isoniazid. Early studies using a 1:3 isoniazid:rifampicin particle ratio found that a subfraction of the formulated microparticles were small enough for inhaled delivery (38% of particles were < 3 µm), and that these particles were readily phagocytosed by cultured murine macrophages [67]. These results were recapitulated in rats following nose-only aerosol exposure. However, issues have been reported surrounding multidrug formulations involving rifampicin, in which combinations of rifampicin with isoniazid and/or pyrazinamide lead to drug breakdown and the formation of adduct-rifamycin isonicotinyl hydrazone [66,67].

The most recent aerosolized antibiotic to enter clinical trials was a dry-powder formulation of capreomycin. Currently, capreomycin treatment in MDR-TB therapy requires daily intramuscular or intravenous injections [68]. Preclinical investigations of inhaled capreomycin in guinea pig models showed lung tissue levels well above MIC values, and increasing drug accumulation in multiple-dose studies [69]. Further, the strategy effectively reduced the bacterial burden in TB infection models, demonstrating superior efficacy over conventional parenteral routes [70]. A Phase 1 trial of dry powder capreomycin indicated that inhaled administration was safe, and no serious or adverse events were observed throughout the study duration in healthy volunteers. Pharmacokinetic analysis showed that a single administration of inhaled capreomycin was able to reach serum concentrations above the MIC (i.e., >2 µg/mL) after administration of 300 mg of the powder [71]. Sputum concentrations were not measured in this study, and no comment could be made with regards to capreomycin exposure within the lungs.

### 3.2. Nontuberculous Mycobacteria

Inhaled amikacin was one of the first reported nebulized therapeutics used for NTM treatment, although administration is currently only recommended if parenteral administration is deemed impractical [23]. Retrospective observational studies have demonstrated some efficacy, as the addition of nebulized amikacin to treatment regimens for refractory NTM-PD has led to persistently negative sputum cultures in 25–43.5% of patients [72,73,74]. Despite the promising results, adverse events were common. Up to 50% of patients developed at least some form of toxicity following treatment [73], and in a second study, 35% of patients had to discontinue therapy due to adverse events [74].

In an effort to recapitulate the positive effects of pulmonary amikacin delivery while mitigating adverse drug events, a liposomal amikacin formulation (Arikayce; Insmed, NJ, USA) was developed. In vitro and in vivo studies demonstrated that this formulation was equally as effective as free amikacin [75], and that it was more readily taken up by alveolar macrophages [76,77]. Clinically, Arikayce has demonstrated statistically significant improvement in bacterial clearance when added to GBT in refractory *M. avium* patients, leading to sputum conversion in 29% of patients compared to 9% of patients on GBT-only [78,79]. These microbiological results were sufficient for Arikayce to receive FDA approval for a subset of patients with limited options after multidrug therapy, becoming the first approved inhaled therapeutic specifically designed for NTM-PD. The clinical and commercial success of Arikayce has provided a benchmark for the potential success of an inhaled therapy in NTM-PD.

## 4. Ongoing Drug Development Programs

The rise of MDR- and XDR-TB, as well as a lack of NTM-specific therapeutics, has reinvigorated the exploration of inhaled therapeutics for mycobacterial lung disease. Below is a list of some of the active drug development programs.

### 4.1. First Line TB Antibiotics

Since the initial reports of rifampicin degradation in fixed-dose combinations, more recent efforts have evaluated different permutations of inhalable first-line TB antibiotics. New fixed dose combinations have been used in an effort to avoid the previously observed degradation, such as a combination of isoniazid and rifabutin microparticles [80]; a triple-combination of isoniazid, rifampicin, and pyrazinamide [81]; and solid lipid microparticles of rifampicin alone [82]. Macrophage uptake studies have demonstrated that the isoniazid and rifabutin particles can be taken up by alveolar macrophages in mouse in vivo models [80], and that rifampicin microparticles are efficiently taken up by murine macrophages in vitro [82]. Aerosol testing of these dry powder programs has demonstrated that all formulations had fine particle fractions and aerosol characteristics suitable for inhaled delivery.

### 4.2. Inhaled Clofazimine

Clofazimine, a rimophenazine dye, is one of the oldest antimicrobials for the treatment of mycobacterial infections [83,84]. Originally developed as a treatment to TB, clofazimine has since been used for the treatment of multibacilliary leprosy [85]. Of particular interest is the ability of clofazimine to localize within macrophages, which suggests an inhaled drug preparation could demonstrate improved antimicrobial activity, without the need for liposomal formulations or other adjuvants [86].

Interest in clofazimine activity against MDR-TB was recently reignited after a series of studies found that the addition of clofazimine to GBT regimens led to a significant improvement in treatment outcomes, and a reduction in the overall duration of therapy [87]. This new regimen successfully prevented relapse for up to two years in patients that had ceased therapy [88]. These studies have paved the way for other investigations of clofazimine addition to GBT for tuberculosis [89,90,91,92,93,94,95] and NTM-PD [96,97,98,99], the majority of which have demonstrated significant clinical or microbiological benefit. However, clofazimine has been shown to have a high incidence of adverse drug events over the duration of therapy (between 76–88% of patients), and is associated with a discontinuation rate of 5–33% [91,97,99,100]. This is likely due to the accumulation of clofazimine following oral administration within the spleen, liver, fat, and gastrointestinal tissues, leading to skin discoloration and gastrointestinal intolerance [101,102].

In an effort to maintain clofazimine’s antimycobacterial activity in the lungs, while minimizing extrapulmonary tissue exposure and adverse drug events, several groups have investigated the use of inhaled clofazimine. Administration of clofazimine dry powder microparticles into the lungs of TB-infected mice demonstrated significant improvement in bacterial clearance relative to oral delivery of similar drug concentrations [103]. Other groups have furthered this strategy by developing novel dry powder formulations for inhalation [104,105].

Most recently, our group has begun the development of an inhaled clofazimine suspension for administration via nebulizer device, due to the limitations of dry powder development described above. This inhalation suspension significantly improved bacterial clearance in NTM-infection models in vivo relative to oral clofazimine, and pulmonary delivery significantly increased the drug concentration within the lung tissues compared to oral administration [106].

### 4.3. Inhaled Azithromycin

Macrolides are critical antibiotics for the treatment of both *M. avium* complex and *M. abscessus* complex pulmonary disease, and are one of the only oral antibiotics with activity against these difficult-to-treat species [107]. Unfortunately, long-term use is often associated with gastrointestinal intolerance, QT elongation, and reversible hearing impairment [108]. Two formulations of inhaled azithromycin have currently been reported—a nebulized formulation and a dry powder. Nebulized azithromycin was shown to produce aerosol particles within a respirable range using multiple different nebulizer devices, which is likely to maximize drug delivery to the lung [109]. Comparatively, dry powder formulations produced less respirable particles than its nebulized counterpart; only 37.5% of the drug was within the respirable fraction following the addition of leucine as a drug carrier [110].

### 4.4. Inhaled Ciprofloxacin

Ciprofloxacin is a quinolone antibiotic currently recommended for the treatment of specific NTM species, such as *Mycobacterium xenopi* and *M. abscessus* (particularly in the case of macrolide-resistant infections) [23]. However, issues with patient tolerability and poor bioavailability have limited ciprofloxacin use. An inhaled liposomal formulation of ciprofloxacin (Linhaliq; Aradigm Corp., Hayward, CA, USA) has already completed both Phase 2 [111] and Phase 3 [112] clinical trials for the treatment of non-cystic fibrosis bronchiectasis associated infections, the results of which support the rationale for the use of inhaled ciprofloxacin.

In preclinical studies of NTM models, both liposomal and nanocrystalline ciprofloxacin formulations were able to penetrate infected macrophages, reduce the growth of *M. avium* and *M. abscessus* in in vitro intracellular infection models, and infiltrate established biofilms to kill *M. avium* and *M. abscessus* [113]. Furthermore, the in vivo efficacy of both liposomal ciprofloxacin and Linhaliq against *M. avium* and *M. abscessus* was demonstrated using mouse models of intranasal infection [113]. In a separate study, daily administration of an inhaled liposomal formulation of ciprofloxacin significantly reduced bacterial recovery in murine models of *M. avium* subsp. *hominissuis* infection [114].

### 4.5. Recombinant Human GM-CSF

Granulocyte-macrophage colony stimulating factor (GM-CSF) is a crucial factor for alveolar macrophage proliferation and immunomodulatory functions [115]. Studies have demonstrated that GM-CSF plays a key role in mediating the host response against *M. abscessus* infection, and that increasing concentrations of GM-CSF in the lung may improve clinical outcomes [116,117,118]. Notably, a reduction in GM-CSF concentrations within cystic fibrosis patients led to the hypothesis that this immunostimulatory factor may be responsible for the diminished immune response and increased susceptibility observed in this population [119]. However, because this is an innate growth factor, administration of exogenous GM-CSF may activate a number of potential extrapulmonary targets, leading to detrimental side effects and whole body inflammation [120]. Thus, administration of GM-CSF may stimulate the pulmonary immune response against mycobacterial infections, and the use of an inhaled preparation is likely to limit drug spillover and extrapulmonary exposure.

Clinically, inhaled GM-CSF has been investigated for the treatment of numerous indications, including pulmonary alveolar proteinosis [121,122,123] and lung metastasis [124]. Molgradex (Savara Pharmaceuticals), a preparation of recombinant human GM-CSF (rhGM-CSF), is used to activate the immune system to respond to infection [115]. In vivo studies have shown that this compound induces the innate and adaptive immune systems through immunostimulatory mechanisms [116,125]. Inhaled rhGM-CSF is currently undergoing Phase 2 (NCT03597347) and Phase 2a (NCT03421743) clinical trials for use against NTM infections and NTM-PD in CF, respectively.

### 4.6. Nitric Oxide Gas

Endogenously, nitric oxide (NO) has numerous biological functions, including acting as a vasodilation signaling molecule and as an antimicrobial compound during innate immune responses [126]. Exogenous nitric oxide gas (gNO) was initially developed as a vasodilator to treat pulmonary hypertension in newborns and improve oxygenation during acute respiratory distress syndrome [127], but more recent studies have shown that gNO has antimicrobial properties at concentrations between 120–200 ppm [128,129], and was able to eradicate *Mycobacterium smegmatis* in vitro after 10 h exposure at 200 ppm [130]. However, because NO is a natural signaling molecule used throughout the body and extrapulmonary exposure can lead to significant adverse events, inhalation is the ideal method of administration [126].

There are currently two groups developing inhalable gNO products. Both of these groups have successfully completed Phase 1 trials, and both have demonstrated the safety and tolerability of intermittent administration of 160 ppm. Notably, administration of 160 ppm gNO, three times daily five times per week in outpatient facilities, led to improvements in bacterial clearance of various bacterial species, including *M. abscessus*, as well as improved pulmonary function for short periods of time following treatment [131,132]. Moreover, a small compassionate use study demonstrated reduction in bacterial load in both patients with no change in FEV_1_ following treatment [133]. Currently, a pair of Phase 2 clinical trials (NCT03331445 and NCT03208764) investigating these gNO formulations are underway.

The use of scaffolding molecules is also being explored to improve gNO delivery and in vivo efficacy at the infection sites, increase the delivered dose, and controlling gaseous release [134,135].

## 5. Regulatory Considerations for Inhaled Drug Programs

Approval of a drug product for marketing requires submitting an application to a governmental regulatory body such as the Food and Drug Administration (FDA) in the United States or European Medicines Agency (EMA). These organizations publish guidance documents that allow sponsors to prepare applications suitable for review. In addition, for drug products which address an unmet medical need in the treatment of a serious or life-threatening condition, programs exist for expedited development and review [136]. A more thorough discussion of the regulatory aspects surrounding the development of inhaled drug products can be found elsewhere [57,137], but a few topics warrant discussion here.

Modern pharmaceutical regulations are intended to ensure marketed drug products are safe and efficacious. Traditional regulatory pathways such as 505(b)(1) and 505(b)(2) in the US require that the sponsor provide evidence of the clinical effectiveness of the drug product, typically data that demonstrates that the drug product has a statistically significant clinical benefit compared to placebo in one or more well controlled clinical trials. This well-intentioned goal can raise significant hurdles in the development of new therapies.

Clinical meaningful endpoints are generally regarded as those that measure how the patient feels, functions, or survives. Measuring how patients feel is typically done through a Patient Reported Outcome questionnaire which is often disease specific, and if not done carefully can introduce significant variability and systematic bias into the trial results [138]. This variability can cause trials to become larger and more expensive. Direct measurements of patient function, such as pulmonary function tests, require strict protocols and consistent effort by patients to minimize variability.

In the case of treating lung infections, eradication of the infection itself is considered a surrogate endpoint, i.e., it is an effect of the treatment that might correlate with a clinical endpoint. Fortunately, there are expedited programs with the FDA such as Breakthrough Therapy which allow for approval based on clinical trials measuring surrogate endpoints [136]. However, the sponsor must provide strong evidence that the surrogate endpoint is predictive of an improved clinical outcome.

## 6. Cross-Fertilization between TB and NTM Development Programs

The rise of MDR-TB has led to a concerted effort to discover and develop novel TB antimicrobials. While this has led to the discovery of novel antibiotics, development programs focusing on NTM-specific therapies are severely lacking. Therefore, in the absence of specific development programs for NTM drug discovery, leveraging TB drugs with broad-spectrum antimycobacterial activity to build the NTM armamentarium may be a successful strategy in the interim, and would also lead to new candidates for inhaled delivery.

Clofazimine is one such example of a potential drug candidate for cross-fertilization, as research programs in recent years have focused on its potential to treat both TB and NTM infections. The benefits of clofazimine in addition to GBT has been thoroughly studied by numerous MDR-TB preclinical and clinical studies [87,139,140,141], and completely separate investigations have evaluated its benefit in cystic fibrosis associated NTM-PD [96,97,99]. Clofazimine has demonstrated synergistic activity in vitro against both TB and NTM with antibiotics such as amikacin and bedaquiline, as well as gNO [142,143,144,145]. Since conventional treatments involve multidrug therapies, the use of drugs with synergistic activity against a breadth mycobacterial species, such as clofazimine, would be beneficial, and worthy of continued evaluation.

Bedaquiline, a new diarylquinolone with a mechanism of action similar to clofazimine, was originally developed for the treatment of MDR-TB, as it has potent antimicrobial activity against a wide range of *M. tuberculosis* strains [146]. This drug is a prime candidate for cross-fertilization into the NTM field, as other studies have shown it has antimicrobial activity against multiple NTM species in vitro [147,148] and in vivo [149,150]. It has also demonstrated synergistic activity in both NTM [144,150] and TB [151] models, which would be of added benefit to both treatment regimens.

Unfortunately, the broad-spectrum activity shown by both clofazimine and bedaquiline is not universal with all anti-TB therapeutics. Many novel agents (such as DC-159a, SQ-641, and tigecycline) have variable activity against mycobacteria at best, while the antimicrobial activity of other compounds appears to be limited to either *M. tuberculosis* (such as delamanid and pretomanid) [152] or NTM (such as clarithromycin and cycloserine) [153,154,155]. Active testing of TB-specific drugs for anti-NTM activity, and vice versa, may provide a wealth of potential therapeutic options for both diseases, but must be approached with some caution.

## 7. Barriers to Entry of Inhaled Therapeutics

In lieu of the studies and positive data presented, there has been minimal development of an inhaled antimicrobial for TB, and only one successful product has been specifically developed for NTM, although development efforts for this disease are relatively new. While there is no clear-cut explanation as to why inhaled therapeutics have not been made clinically available for TB, there are a number of potential reasons.

One of the major barriers to any novel TB-specific therapeutic is the economic limitations of developing countries, which have substantially higher incidences of TB infection than wealthier nations [5]. This forces treatment regimens, and by extension any new therapeutic, to be as inexpensive as possible, in order to be as accessible as possible. The added cost of inhalation devices and formulation development often involved in inhaled therapeutics may be perceived as too expensive for developing regions, particularly when compared to “cheaper” oral tablets. These oral formulations are often seen as “effective enough”, despite the limitations of systemic drug administration, such as non-specific tissue distribution [101,102], high incidence of drug–drug interactions [156], or the fact that a number of these systemic antibiotics are unable to adequately penetrate the granulomatous structures [157,158].

In the treatment of tuberculosis in developing countries, there must be specific logistic considerations surrounding inhalation device use that are different from those in western countries. Limitations in available medical staff, and the distance between patients and medical facilities may also hamper the use of inhaled antibiotics. The ability to teach patients proper inhalation procedures and patient monitoring may be severely limited in these regions, leading to poor technique, poor compliance, and overall treatment failure. These factors must be considered during the product use in these areas, in order to minimize risk of developing drug resistance.

Similar to streptomycin-resistant TB in the 1950s and 1960s, the potential benefits of inhaled therapeutics has recently been outshined by the discovery of new antibiotics, such as bedaquiline [146], delamanid [159], and pretomanid [160]. These medications have been moved along the development pipeline despite concerns of adverse events (i.e., QTc elongation) and efficacy [161,162]. Even more concerning is the evidence of antibiotic resistance that is already being observed in clinical isolates [163]. While the importance of these new antibiotics for improving treatments against MDR- and XDR-TB cannot be overstated, the role, and benefit, of inhaled antimicrobial agents should not be overlooked again. One potential solution of to all issues may be the use of these novel drugs as inhaled formulations. The combined approach would further contribute to their benefit for MDR-TB patients by limiting the toxicity observed through oral and parenteral administration, while high site concentrations would prevent the development of resistance, increasing the overall drug lifespan.

Lastly, the lack of clinical efficacy data surrounding inhaled therapeutics in TB has limited the development of inhaled antimicrobials. The most recent inhaled therapy to enter clinical trials, capreomycin, demonstrated safety, but has not been able to advance into clinical studies that would directly compare the efficacy of inhaled therapies to oral and/or parenteral treatments [71]. This positive feedback loop—a lack of inhaled products for TB prevents clinical studies, which prevents inhaled products for TB—is a major limiting step in the development of inhaled products. However, the recent success of Arikayce against NTM may act as a pioneer, as it is the most recent therapeutic to demonstrate benefit in mycobacterial infections. The only hope is that the success of liposomal amikacin can provide a rationale for all inhaled antimycobacterial therapies, thereby increasing the number of clinical studies performed and subsequently the benefits of inhaled antimicrobials to these chronic, difficult to treat pulmonary infections.

## 8. Conclusions

Mycobacterial lung diseases are difficult-to-treat infections that require extensive multidrug treatment strategies and are often marred by the high incidence of treatment emergent adverse events due to extensive systemic drug administration. Given the pulmonary-specific nature of these infections, it is worth considering the benefits of inhaled anti-infective therapy, i.e., the high local concentration at the site of infection, local penetration into the infected lung tissue, and, most importantly, reduced systemic levels and side effects. Both preclinical and clinical studies have repeatedly demonstrated that an inhaled therapy, complementary to current GBT strategies, has the potential to be clinically beneficial for all types of mycobacterial infections. Continued research, and open-minded evaluations of this important strategy should be carried out to continue investigating their potential addition to the antimycobacterial therapeutic arsenal.

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
