# Peer review of "Inhaled Antibiotics for Mycobacterial Lung Disease"

_pharmaceutics, 2019, doi:10.3390/pharmaceutics11070352_

Round 1

Reviewer 1 Report

The manuscript is well written and organized an it is worth to be published after some revision.

Although generally clear and nicely discussing some of the main aspects restraining the development of inhaled antibiotic therapies, there are some other important issues that require attention such as regulations and poor clinical settings and diagnostic tools. Moreover, the evolution of disposable DPIs and nebulizers has not been considered which in some cases may counterbalance the higher costs compared to other conventional medications and increase cost-efficacy of inhaled therapies. 

I suggest the authors to consider the addition of some critical discussion on these points before publication 

Author Response

Dear Reviewer,

I would like to thank you for your time and careful review of this article. We have addressed your comments below; wherein the reviewer comments have been bolded and our response to each point immediately follows the associated comment.

Although generally clear and nicely discussing some of the main aspects restraining the development of inhaled antibiotic therapies, there are some other important issues that require attention such as regulations and poor clinical settings and diagnostic tools

We agree with you that regulations, poor clinical settings, and diagnostic tools are important topics to be considered in terms of mycobacterial lung diseases and their treatment. As such, we have included brief comments on both the regulatory aspects (Lines 360 - 387) of inhaled drug development and issues related to the clinical settings of treatment (Lines 437 - 443).

With that being said, we believe that thorough discussion of these issues is beyond the scope of this review article, which is intended to discuss inhaled antibiotics and their potential role in the treatment of mycobacterial lung diseases. We believe that a separate review is required to comprehensively discuss these topics.

Moreover, the evolution of disposable DPIs and nebulizers has not been considered which in some cases may counterbalance the higher costs compared to other conventional medications and increase cost-efficacy of inhaled therapies.

We would like to thank you for this comment, as it brought to our attention that inhaled drug delivery devices were generally overlooked in this review article. As such, we have added a section describing the different nebulizer devices and the pros/cons of drug development and the administration methods of these specific devices (Section 2).

Reviewer 2 Report

Dear Associate Editor,

With great pleasure I have evaluated the review article "Inhaled antibiotics of mycobacterial lung disease" by Banaschewski and Hofmann.

This comprehensive narrative review is well written and enjoyable to read. There are only minor issues to address before the article is suitable for publication.

Minor suggestions for improvement:

Introduction. Given the comprehensive nature of the review references on the economic burden and mortality of NTM-PD as well as on treatment outcomes in MAC-PD and non-MAC NTM-PD should be included in the introductory section.

Introduction. Line 78: add "culture" to "... sputum conversion."

Introduction: Lines 82ff: treatment discontinuation rates of 9-19% for patients with NTM disease are an underestimation, in particular given a very common reason for discontinuation: the poor treatment success after a sufficient treatment period (see systematic reivew above). Add myelotoxicity to "common adverse events".

Section 1 and 2: Lines 89 and 138: use "adherence" instead of "compliance".

Introduction: Line 94: shoud read "This targeted delivery has been ...".

Section 3. Line 243: provide full details for reference 95.

Section 3. Line 247: macrolides are the most important antibiotic for MAC-PD as well.

Section 3. Lines 259ff: provide references for the mentioned phase 2 and 3 clinical trials (or NCT identifier, if appropriate).

Section 3. Lines 291-292: provide NCT identifier for the mentioned clinical trials.

Section 3. Line 312: provide NCT identifier for the mentioned clinical trial.

Section 5. Lines 376-378: provide NCT identifier for the mentioned clinical trial on capreomycin.   

Author Response

Dear Reviewer, 

I would like to thank you for your time and careful review of this article. We have addressed your comments below; wherein the reviewer comments have been bolded and our response to each point immediately follows the associated comment.

Introduction. Given the comprehensive nature of the review references on the economic burden and mortality of NTM-PD as well as on treatment outcomes in MAC-PD and non-MAC NTM-PD should be included in the introductory section.

Thank you for this point. I have added additional information regarding the estimated economic burden (Lines 91 - 94), and supplemented the discussion on the variability in treatment outcomes (Lines 83 - 87). In addition, we've also added a brief discussion on the differences in mortality following infection with MAC-PD, compared to other NTM species (Lines 87 - 90)

Introduction. Line 78: add "culture" to "... sputum conversion."

This correction has been addressed

Introduction: Lines 82ff: treatment discontinuation rates of 9-19% for patients with NTM disease are an underestimation, in particular given a very common reason for discontinuation: the poor treatment success after a sufficient treatment period (see systematic reivew above). Add myelotoxicity to "common adverse events".

I am not sure exactly what you're referring to in your comment "See systematic review above" (I may be missing something?), but have attempted rectify these discontinuation rates. Reviewing articles by Fields et al. and Xu et al., I have adjusted the report to now read "9 - 39%", to reflect the highest discontinuation rate reported by Xu et al, and added a comment on lack of observable benefit being responsible (in part) for discontinuation. I hope this addresses your concerns.

Additionally, I added myelotoxicity to the common adverse events, as requested.

Section 1 and 2: Lines 89 and 138: use "adherence" instead of "compliance".

These have been addressed, and changed.

Introduction: Line 94: shoud read "This targeted delivery has been ...".

This typo has been corrected.

Section 3. Line 243: provide full details for reference 95.

The DOI has been added to the reference. Since the article is still in press at this time, no other details are available as of yet.

Section 3. Line 247: macrolides are the most important antibiotic for MAC-PD as well.

You are correct, and this point has been added to Line 288 (formerly line 247).

Section 3. Lines 259ff: provide references for the mentioned phase 2 and 3 clinical trials (or NCT identifier, if appropriate).

Citations for these two clinical trials (reference 110 and 111) have been added to Line 305.

Section 3. Lines 291-292: provide NCT identifier for the mentioned clinical trials.

The NCT identifiers have been added to Line 334.

Section 3. Line 312: provide NCT identifier for the mentioned clinical trial.

The NCT identifiers have been added to Line 355.

Section 5. Lines 376-378: provide NCT identifier for the mentioned clinical trial on capreomycin.

Unfortunately, I was unable to find the appropriate NCT identifier for this trial. However, the publication citation has now been added, which describes the trial methods and results (Line 457). I hope this will suffice.

Round 2

Reviewer 1 Report

The manuscript is acceptable for publication